# WHAT'S WRONG WITH THE ROBUSTNESS OF OBJECT DETECTORS?

## ABSTRACT

Despite tremendous successes achieved, object detection models confront the vulnerability to adversarial attacks. Even with imperceptible adversarial perturbations in images, they probably yield erroneous detection predictions, posing a threat to various realistic applications, e.g., medical diagnosis and automatic driving. Although some existing methods can improve the adversarial robustness of detectors, they still suffer from the detection robustness bottleneck: the significant performance degradation on clean images and the limited robustness on adversarial images. In this paper, we conduct empirically a comprehensive investigation on *what's wrong with the robustness of object detectors in four different seminal architectures*, i.e., two-stage, one-stage, anchor-free, and Transformer-based detectors, inspiring more research interest on this task. We also devise a *Detection Confusion Matrix* (DCM) and *Classification-Ablative Validation* (ClsAVal) for further detection robustness analyses. We explore underlying factors that account for robustness bottleneck. It is empirically demonstrated that robust detectors have reliable localization robustness and poor classification robustness. The classification module easily mis-classifies the foreground objects into the background. Furthermore, Robust Derformable-DETR suffers from a poor classification and localization robustness. Our source codes, trained models, and detailed experiment results will be publicly available.

## 1 INTRODUCTION

With numerous breakthroughs in recent years, deep neural networks (DNNs) have built a series of milestones in the computer vision community He et al. (2016); Chen et al. (2018a); Ge et al. (2021). Nevertheless, with millions of model parameters, they are verified to be easily fooled to generate completely wrong predictions under slight and imperceptible image perturbations Szegedy et al. (2014). Many recent works are devoted to exploring the model robustness with adversarial perturbations crafted by attack models including FGSM Goodfellow et al. (2015), PGD Madry et al. (2018), AdvGAN Xiao et al. (2018), Carlini and Wagner Attack (C&W) Carlini & Wagner (2017). However, those mainly focus on the image classification task. The robustness of object detectors is quite under-explored.

Object detectors not only identify which categories objects belong to (*classification*), but also recognize where objects exactly are (*localization*). Inevitably, detection models suffer from more complex and challenging adversarial robustness from classification and localization, exposing more possibilities of being attacked. A few recent approaches Xie et al. (2017); Wei et al. (2019); Sarkar et al. (2017); Chen et al. (2018b) have been proposed to attack the state-of-the-art object detectors. Though those methods achieve successful attacks on Faster RCNN and SSD object detectors, it is not trivial to defense attacks to ensure the adversarial robustness for object detectors. Generally, few works are devoted to adversarially-robust object detectors. Three early attempts are MTD Zhang & Wang (2019), CAWT Chen et al. (2021) and RobustDet Dong et al. (2022), which are derived from the one-stage SSD detector.

Nevertheless, existing robust detection methods present a typical **detection robustness bottleneck** on both clean images and adversarial images for object detection: *a significant performance decline on clean images and the limited performance on adversarial images.* As demonstrated in Fig. 1, on one hand, robust object detectors (MTD and CWAT) only obtain about **22%~37% mAP on adversarial images!** On the other hand, robust object detectors (MTD and CWAT) suffer from the

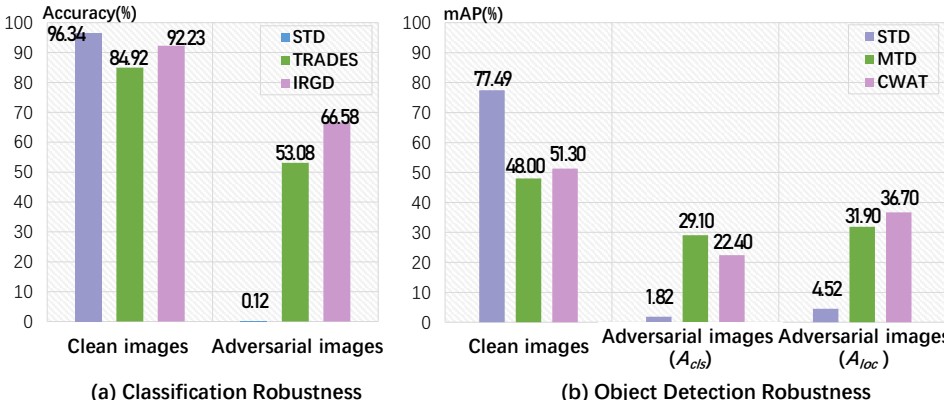

Figure 1: Robustness for image classification models and object detection models. "STD" is the standard model (non-robust), *i.e.*, ResNet in classification and SSD in detection. $A_{cls}$ and $A_{loc}$ denote the attacks for classification and localization in detection.

performance **decrease by nearly 30% mAP on clean images** (77.49% mAP for standard SSD *vs.* 48% mAP for MTD, 51.3% mAP for CWAT)! Instead, this is not an intuitive phenomenon, because the robust classification models (*e.g.*, TRADES Zhang et al. (2019), IRGD Gowal et al. (2021)) have a small amount of the performance decline on clean images while gaining the robustness, as shown in Fig. 1. A same thing for those classification and detection methods is adversarial training to ensure the robustness, but this problem occurs notably in object detection. Therefore, it is worthy of more attention to exploring what's wrong with the robustness of object detectors.

In this paper, rather than proposing a specific solution for robust object detectors as usual, we specifically aim to make an initial attempt to investigate the robustness of object detectors systematically and comprehensively as possible as we can, paving a way for future works that are inspired to essentially mitigate the detection robustness bottleneck. The involved object detectors cover **four seminal object detectors** with different structures, *i.e.*, *the two-stage, one-stage and anchor-free and Transformer-based detectors*. Specifically, we devise a novel *Detection Confusion Matrix* (DCM) in object detection to analyze the classification and localization in detail and *Classification-Ablative Validation* (ClsAVal) for detection robustness analysis. Four robust object detectors with adversarial training present the similar robustness bottleneck aforementioned, which is mainly attributed to the inferior classification robustness, *i.e.*, the mis-classification of the foreground as the background in the classification module, with less confusion among foreground categories. Besides, in Deformable-DETR, both the robustness of classification and localization is poor, unlike SSD, Faster RCNN, and YOLOX with reliable localization robustness and poor classification robustness.

## 2 ADVERSARIAL ROBUSTNESS OF OBJECT DETECTORS

With the remarkable learning capability of deep neural networks, three representative series of deep learning based object detectors are prevalent, leading the recent research on object detection. They are *two-stage*, *one-stage*, *anchor-free* and *Transformer-based* object detectors. In our work, we will empirically investigate the adversarial robustness of those three types of detectors. Concretely, Faster RCNN Ren et al. (2015) (*two-stage*), SSD Liu et al. (2016) (*one-stage*), YOLOX Ge et al. (2021) (*anchor-free*) and Deformable-DETR Zhu et al. (2021) (*Transformer-based*) are selected. In this section, adversarial training for robust object detection will be firstly described and then empirical analyses on detection robustness will be elaborated. For SSD, Faster RCNN and YOLOX, we use PASCAL VOC Everingham et al. (2015) dataset for analysis. For Deformable-DETR, MS-COCO Lin et al. (2014) is adopted, since it is unable to successfully train Deformable-DETR on PASCAL VOC using the official code. (However, we still provide successful training results on VOC dataset using MMDetection codes in the supplementary material.)

### 2.1 PRELIMINARIES

Object detection can be regarded as a multi-task learning for classification and localization. Formally, a detection model $f$ is parameterized by $\boldsymbol{\theta}$, which consists of a backbone $f_b$ and two headers of classification $H_{cls}$ and localization $H_{loc}$. Given an input image $x \in \mathcal{D}$, two headers yield probabilistic confidence and the predicted localization for each bounding box, respectively. After Non-Maximum

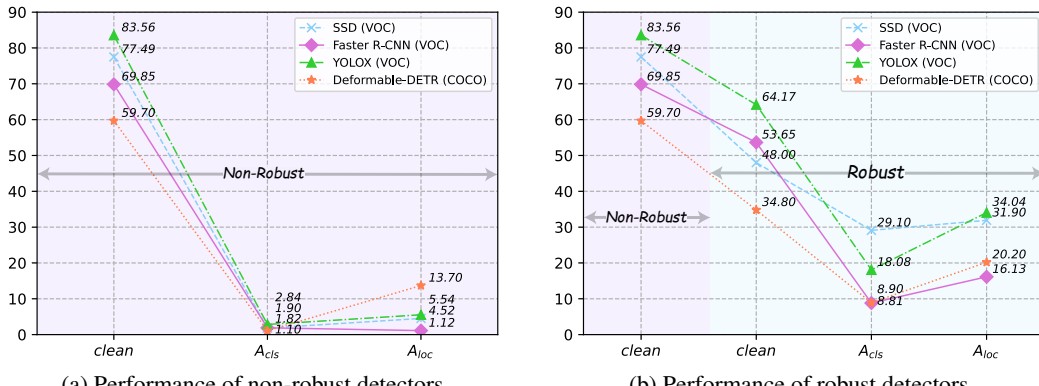

(a) Performance of non-robust detectors        (b) Performance of robust detectors

Figure 2: Performance of non-robust and robust detectors with different architectures on clean images and adversarial images. (Best viewed zoomed in color.)

Suppression (NMS) and filtering[1], the object detector $f$ outputs $N$ detected bounding boxes $\hat{\boldsymbol{b}}_k$ ($k = 1, ..., N$) with the confidence $\boldsymbol{s}_k \in \mathbb{R}^{C+1}$ over $C$ object categories and the background. In the training phase, the objective function integrates the classification loss $\mathcal{L}_{cls}$ and localization loss $\mathcal{L}_{loc}$ to be optimizedly minimized:

$$\min_{\theta} \mathbb{E}_{(x,\{\boldsymbol{y_i},\boldsymbol{b_i}\})\sim\mathcal{D}} \mathcal{L}_{cls}(\boldsymbol{s} = H_{cls}(f_b(x)), \{\boldsymbol{y_i},\boldsymbol{b_i}\}) + \mathcal{L}_{loc}(\hat{\boldsymbol{b}} = H_{loc}(f_b(x)), \{\boldsymbol{y_i},\boldsymbol{b_i}\}), \quad (1)$$

where $y_i$ is the category label of Ground-Truth (GT) bounding box $\boldsymbol{b}_i$.

Existing attack methods for object detectors are essentially cast as attacking by using variants of individual classification or localization task losses or their combinations Xie et al. (2017); Wei et al. (2019); Liu et al. (2019); Chen et al. (2018b). Following those detection attack works, we denote the attack against the classification as $A_{cls}$ and the attack against the localization as $A_{loc}$, respectively.

$$A_{cls}(x) = \arg\max_{\bar{x}\in\mathcal{S}_x} \mathcal{L}_{cls}(H_{cls}(f_b(\bar{x})), \{\boldsymbol{y_i},\boldsymbol{b_i}\}),$$
$$A_{loc}(x) = \arg\max_{\bar{x}\in\mathcal{S}_x} \mathcal{L}_{loc}(H_{loc}(f_b(\bar{x})), \{\boldsymbol{y_i},\boldsymbol{b_i}\}), \quad (2)$$

where $\bar{x}$ is the adversarial image of $x$, and $\mathcal{S}_x = \left\{\bar{x} \cap [0, 255]^{cwh} \mid \|\bar{x} - x\|_{\infty} \leq \epsilon\right\}$ is the adversarial image space centered on clean images $x$ with perturbation budget of $\epsilon = 8$. Accordingly, the objective function for robust object detection is formulated as

$$\min_{\theta} \mathbb{E}_{(x,\{\boldsymbol{y_i},\boldsymbol{b_i}\})\sim\mathcal{D}} \mathcal{L}_{cls}(H_{cls}(f_b(A(x))), \{\boldsymbol{y_i},\boldsymbol{b_i}\}) + \mathcal{L}_{loc}(H_{loc}(f_b(A(x))), \{\boldsymbol{y_i},\boldsymbol{b_i}\}), \quad (3)$$

where $A$ represents the attack in the adversarial training. Ideally, a robust object detector should not only have promising localization robustness and classification robustness, but also present a remarkable performance on clean images.

## 2.2 EMPIRICAL ANALYSES ON DETECTION ROBUSTNESS

In this section, we empirically explore the adversarial robustness for four series of state-of-the-art detection models. Our attack in the adversarial training involves simultaneously classification and localization attacks in this paper, *e.g.*, MTD Zhang & Wang (2019). We adopt the similar experimental setup of attacks and criteria as MTD. We evaluate detection models under the attack $L_{cls}$ and $L_{loc}$ using the PGD-20 ($\epsilon = 8$), and use the mean Average Precision (mAP) as the robustness evaluation metric. The performance evaluation on the non-robust and robust models with different detection structures is shown in Fig. 2.

*1) Non-robust models on clean and adversarial images:* The standard non-robust models have a remarkable detection performance over 60% mAP on clean images. But, all of them have extremely poor results on adversarial attacks which is lower than 10% mAP.

*2) Robust models on clean and adversarial images:* Robust detectors have limited robustness to adversarial attacks achieving lower than 40% mAP in the best case and only 10% in the worst case. Meanwhile, they have a performance decrease on clean images compared with non-robust models.

*3) Robust models vs. non-robust models on clean images:* Compared with the non-robust model, robust detectors suffer from a significant degradation by 15% mAP drops on average, and even by up to 30% at most. Deformable-DETR also has about 25% mAP drop on COCO.

---

[1]It specifically represents the process of filtering bounding boxes whose scores are large than the threshold for final detection results in detectors.

Figure 3: Illustration of Detection Confusion Matrix.

*4) Robust models **vs.** non-robust models on adversarial images:* Robust models can maintain a certain detection ability on adversarial images which has about 20% mAP improvement, while non-robust models hardly detect correctly objects on adversarial images (lower than 10% mAP).

*5) Robust models resistant to classification attack and localization attack:* The performance on $A_{cls}$ adversarial image is generally lower than those of $A_{loc}$, indicating the robustness of the classification module is commonly weaker than that of the localization module. Compared with Faster R-CNN, SSD has a smaller robustness gap between the classification and localization modules. The performance of YOLOX on clean images is better than SSD, but its robustness on the classification module is weaker than that of SSD. The robustness gap of YOLOX between the classification and localization module is similar to the two-stage detector.

## 3 WHAT'S WRONG WITH THE ROBUSTNESS OF OBJECT DETECTORS?

In this section, we first devise *Detection Confusion Matrix* and *Classification-Ablative Validation* in object detection for detailed analyses, and then mainly analyze the robustness in classification and localization, taking one-stage SSD for example.

### 3.1 PRELIMINARIES

**Detection Confusion Matrix (DCM).** Considering that adversarial images would be easily misidentified as any other category, it matters that we figure out what the detection confusion is among different object categories due to adversarial attacks. Accordingly, we resort to the confusion matrix which intuitively indicates the detection performance for each category and the confusion between different categories. However, what is responsible for the detection performance evaluation involves classification categories, classification confidences, and the regression localization of bounding boxes. Thus, the conventional confusion matrix for classification improperly suits object detection. In our work, we devise *Detection Confusion Matrix* to diagnose object detection, which indicates the confusion of detection results among different categories and their localization.

Those detected bounding boxes after Non-maximum Suppression (NMS) are considered for calculating DCM. Those whose IOU with all the GTs is lower than $T_{iou}$ are regarded as its true category being assigned to *background*. Otherwise, the true category of a detected bounding box $b_i$ can be assigned to a certain object category. $T_{iou}$ is the IOU acceptable threshold (it is 0.5 in this paper). Once the matching between GT and the predicted bounding box is determined, we can calculate DCM via this information, shown in Fig. 3.

Specifically, a detection confusion matrix $\mathcal{M}$ has the size of $(1 + C) \times (1 + C)$, whose items are named *cells*. In each cell, it simultaneously contains *the category confusion* and *the IOU distribution*. **1)** For the category confusion, in a cell $(m,n)$, $\mathcal{M}_{m,n}^{conf}$ represents the sum of confidences of the bounding boxes whose true category is $c_m$ but predicted to be category $c_n$. As per routine of mAP, we assign the true bounding box and the true category to the predicted bounding box based on its IOU value with GT boxes. All the GT bounding boxes in an image are denoted as $\mathcal{B}$ and all the predicted bounding boxes as $\hat{\mathcal{B}}$. $b_i \in \mathcal{B}$ is the GT bounding box with the largest IOU with a predicted bounding box $\hat{b}_k \in \hat{\mathcal{B}}$. If $IOU(\hat{b}_k, b_i) >= T_{iou}$, then $\hat{b}_k$ matches with $b_i$ and the true category of $\hat{b}_k$ is also the category of $b_i$; otherwise, $\hat{b}_k$ does not match any GT bounding box and its true category is $c_{bg}$. Thus the set of all predicted bounding boxes with the true category $c_m$ can be indicated as: $\hat{\mathcal{B}}(c_m)$, then the category confusion is defined as: $\mathcal{M}_{m,n}^{conf} = \sum_{\hat{b}_k \in \hat{\mathcal{B}}(c_m, c_n)} s_k$, where $\hat{\mathcal{B}}(c_m, c_n) \subset \hat{\mathcal{B}}(c_m)$ is the subset of $\hat{\mathcal{B}}(c_m)$ whose predicted category is $c_n$ but the true category is $c_m$.

**2)** For the IOU distribution, in a cell $(m,n)$, $\mathcal{M}_{m,n}^{iou}$ represents the distribution of IOU values for the bounding boxes whose true category is $c_m$ but predicted to $c_n$. In this distribution, $x$-axis is the IOU value, and $y$-axis is the percentage of the number of bounding boxes in the corresponding interval. It

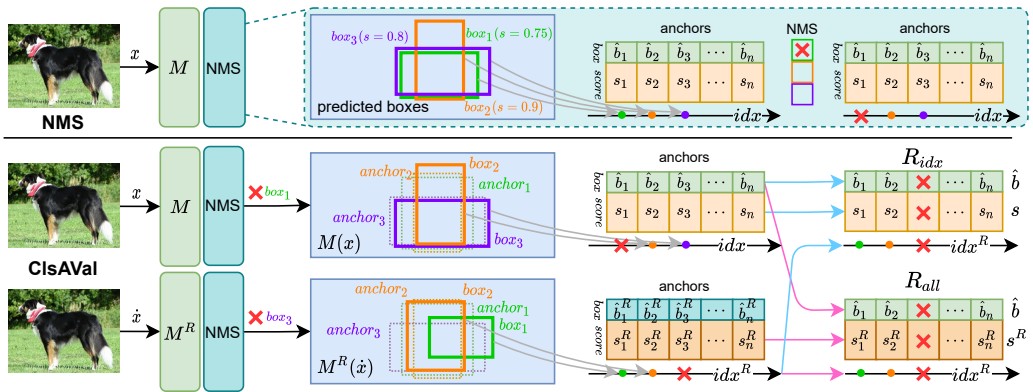

Figure 4: Classification-Ablative Validation. Based on the conventional detection process, the output of the current model classification module is replaced with the output of a reliable model classification module, to eliminate the impact of the classification module and evaluate the localization module performance individually.

is defined as $\mathcal{M}_{m,n}^{iou} = hist_{\hat{\boldsymbol{b}}_k \in \hat{\mathcal{B}}(c_m, c_n)}(IOU(\hat{\boldsymbol{b}}_k, \boldsymbol{b}_i))$, where $\boldsymbol{b}_i$ is the GT bounding box matching $\hat{\boldsymbol{b}}_k$ and $hist(\cdot)$ is the histogram operation.

**Classification-Ablative Validation (ClsAVal).** To independently evaluate the robustness of the localization module, we propose the ClsAVal method. To eliminate the impact of the classification module on the results, it assumes that each proposal has a reliable classification score (this can be provided by a reliable detector).

Given a detection model $M$, its detected object set is $\mathcal{O}(x)$. After applying NMS and filtering, the reserved boxes with their indexes $idx$ will be recorded: $\tilde{\mathcal{O}}(x) = \{(\hat{\boldsymbol{b}}_k, \boldsymbol{s}_k), k \in idx\}$. To diagnose the localization robustness of $M$, it resorts to another model $M^R$, whose performance on $\dot{x}$ is better than $M$ on $x$ and its detected object set is $\tilde{\mathcal{O}}^R(x) = \{(\hat{\boldsymbol{b}}_k^R, \boldsymbol{s}_k^R), k \in idx^R\}$, where $x$ and $\dot{x}$ are from the same image but with or without certain manipulation (*e.g.* adversarial perturbation). Two validation modes are defined, shown in Fig. 4. **1) ClsAVal $\boldsymbol{R}_{idx}$:** directly use the indexes $idx^R$ of $\dot{x}$ to select the bounding boxes on $x$: $\tilde{\mathcal{O}}_{\boldsymbol{R}_{idx}}^R(x) = \{(\hat{\boldsymbol{b}}_k, \boldsymbol{s}_k) | k \in idx^R\}$. At this time, the selected boxes are assigned with correct indexes for positions, but their scores are still from $M$. This indicates what scores $M$ predicts for the boxes in those correct positions. **2) ClsAVal $\boldsymbol{R}_{all}$:** If the selected boxes are assigned with correct indexes for positions and meanwhile their scores are also from $M^R$ on $\dot{x}$, then the selected bounding boxes on $x$ are $\tilde{\mathcal{O}}_{\boldsymbol{R}_{all}}^R(x) = \{(\hat{\boldsymbol{b}}_k, \boldsymbol{s}_k^R) | k \in idx^R\}$, to eliminate the interference of the classification module. **ClsAVal $\boldsymbol{R}_{all}$** reflects the localization predicted by $M$ for those that are at correct positions. Suppose $\dot{x}$ and $M^R$ are the clean image and the standard model, respectively; $x$ and $M$ are their adversarial and robust counterparts, respectively. If $M$ produces similar boxes to $M^R$, it means that $M$ is robust in localization; otherwise, it is non-robust in localization.

### 3.2 ROBUSTNESS IN CLASSIFICATION AND LOCALIZATION

As demonstrated in Sec. 2.2, *robust detection models have limited robustness on adversarial images, and degenerate significantly on clean images.* That is, existing robust object detectors often have an unsatisfactory performance on both adversarial and clean images. One reason is likely that robust models have difficulties to collaboratively learning clean images and adversarial images together. That indicates that those models have serious conflicts between learning clean images and adversarial images in the adversarial training. Thus, they have to compromise a trade-off between the accuracy (on clean images) and adversarial robustness (on adversarial images), leading to the performance sacrifice, especially on clean images. We will elaborately investigate this problem in the following.

**(a) Are robust detectors really poor in localization?**
Compared with image classification, object detection models has one more regression branch for localizing objects. Accordingly, a problem arises: *does the conflict in object detection mainly exist in the regression subtask for object localization?* We will first explore whether the localization is the "chief criminal" that is responsible for the poor robustness in detection. Accordingly, it is necessary to eliminate the interference of the classification module to verify the robustness of the localization module independently. Two possible cases can be considered to explain this issue.

*1) The error of the localization result may be due to the poor robustness of the **localization module itself**, which gives the wrong bounding boxes.*

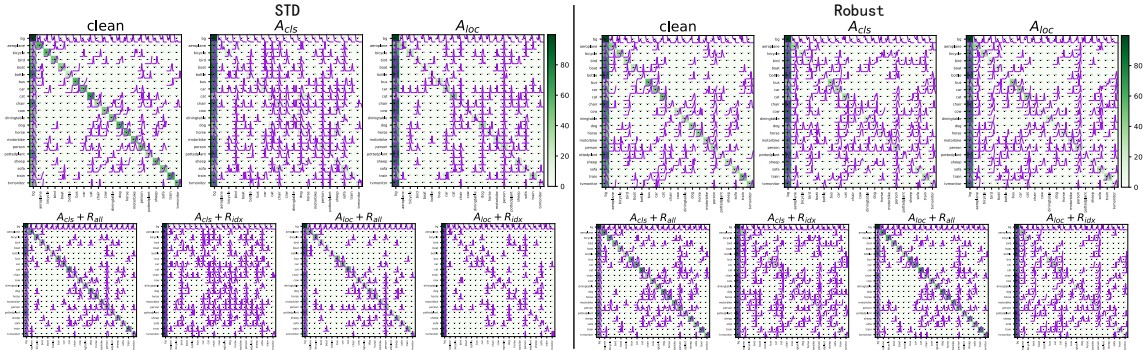

Figure 5: Robustness analyses with *Detection Confusion Matrix* for standard and robust SSD models[2].

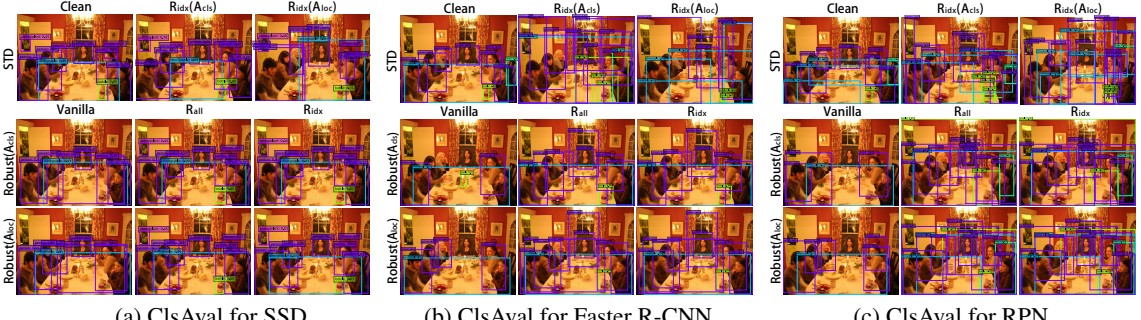

(a) ClsAval for SSD  (b) ClsAval for Faster R-CNN  (c) ClsAval for RPN

Figure 6: (a)(b): *Classification-Ablative Validation* for standard and robust SSD and Faster R-CNN models. "Vanilla" means the standard outputs of the model. (c): *Classification-Ablative Validation* for RPN in standard and robust Faster R-CNN.

*2) The **classification module** yields wrong scores so that the NMS module selects the bounding boxes at wrong positions that do not contain correct objects.*

In order to figure out which factor dominates the error of prediction results, we verify the localization robustness via the proposed ClsAVal. This has a premise that standard detectors can select bounding boxes at correct positions on clean images. As shown in Fig. 6(a), the performance of robust SSD under adversarial attacks is limited: not only the confidence for many objects are quite low, but the bounding box also has a large deviation. However, via $R_{idx}$ validation, under either classification attack $A_{cls}$ or localization attack $A_{loc}$, those boxes (produced by robust models) at correct positions do not deviate much from the detected boxes on clean images by standard SSD. But the confidences for these bounding boxes that should be correct are extremely low and thus wrongly recognized as the background. Compared with $R_{idx}$ validation results of the standard SSD model under $A_{loc}$, it can be seen that the localization module of the detector has indeed gained a good robustness through adversarial training. It means that *the model is not due to the poor robustness of the localization module itself, but **the classification module with poor robustness that gives the wrong confidence of the bounding box**, which makes the subsequent module select wrong bounding boxes.*

**(b) What's wrong with the classification robustness?**
To analyze the classification robustness, we employ the Detection Confusion Matrix to inspect the detection results of robust models. This aims to not only summarize the classification and localization performance for each category but also analyze how confused the detector is for classification and localization among all the categories. Fig. 5 prodives DCM results for standard SSD and robust SSD.

*1) The adversarial attack enforces the standard detection model to easily mis-classify foreground objects into the background.*

*2) The robust model trained by adversarial training also tends to mis-classify foreground objects into the background for detection on both clean images and adversarial images.*

In Fig. 5, compared to the results of the standard model on the clean images, standard and robust models on $A_{cls}$ and $A_{loc}$ adversarial images have a large number of bounding boxes that are incorrectly predicted into the background. Besides, robust models on clean images also have similar results. This reasonably explains that the insufficient classification robustness is mainly attributed to the prediction

---

[2]Best viewed zoomed in color. And we also provide clearer versions of Fig.5-8 in our supplementary.

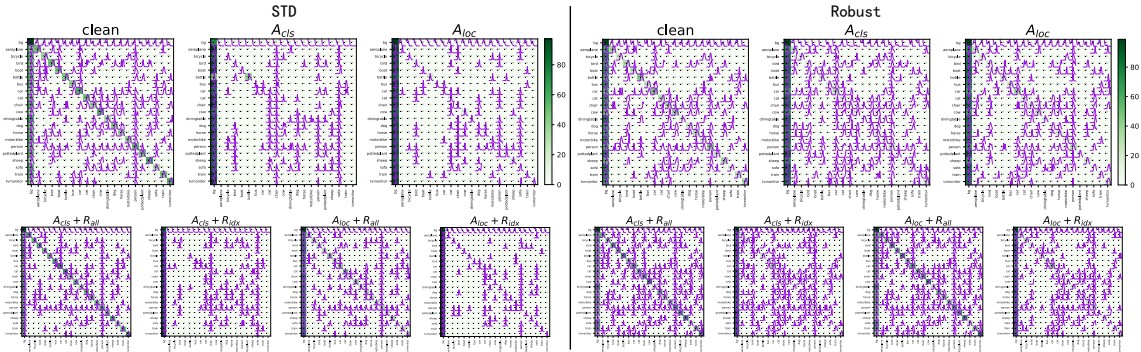

Figure 7: Robustness analyses with *Detection Confusion Matrix* for standard and robust Faster RCNN.

confusion between foregrounds and the background. Furthermore, *these correctly classified objects have larger IOUs, demonstrating that the localization in detectors has higher robustness to some extent.*

*3) The poor performance of robust models is due to the mis-classification of the classification module, which leads to selecting bounding boxes at wrong positions, rather than the poor robustness of the localization model itself.*

In Fig. 5, under $R_{all}$, the standard SSD model and robust SSD are enable to locate objects on $A_{cls}$ or $A_{loc}$ adversarial images, which is similar to the results on clean images. Under $R_{idx}$, robust SSD yields low confidences for those boxes that should be correct, but it is still better than the standard model. Despite the limited robustness, its ability to distinguish objects in the classification module has deteriorated. Overall, it demonstrates that both standard and robust SSD present a reliable localization robustness on adversarial images; however, the conflict of learning together clean and adversarial images seriously affects the classification robustness, which leads to the mis-classification of the foreground into the background.

# 4 ROBUSTNESS IN DETECTORS WITH DIFFERENT ARCHITECTURES

In this section, we will further explore the robustness bottleneck for other three detectors with different architectures including *two-stage* Faster-RCNN, *anchor-free* YOLOX and *Transformer-based* Deformable-DETR.

## 4.1 ROBUSTNESS IN TWO-STAGE DETECTOR

Two-stage detectors require NMS for both stages to filter the detection bounding boxes based on the prediction results of their own classification modules. For the the ClsAVal validation, the first stage uses $R_{all}$ validations. $R_{all}$ and $R_{idx}$ are adopted for the second stage.

**Robustness Analyses on Faster RCNN.** As demonstrated by $R_{idx}$ validation in Fig. 6, standard Faster RCNN cannot accurately detect objects with adversarial perturbations even if the correct bounding box is given. This can draw a conclusion that Faster RCNN itself has poor localization and classification robustness, unlike the standard SSD that presents good localization robustness even without adversarial training. Robust Faster RCNN also suffers from the robustness bottleneck. But, the locations of detected objects highly accurately for both attacks of $A_{cls}$ or $A_{loc}$, demonstrated by $R_{idx}$ and $R_{all}$ in Fig. 6.

With DCMs under either the attack of $A_{cls}$ or $A_{loc}$ in Fig. 7, standard and robust Faster RCNN models mis-classify many foreground objects as the background. From the DCM of $R_{all}$ validation, the robust model shows remarkable performance no matter what kind of attack is used. This further supports the conclusion that *the decline in model performance is not due to the poor robustness of the localization module itself.* From the DCM of $R_{idx}$ validation, *Faster RCNN also incorrectly classifies most of the correct bounding boxes as the background.* Thus, two-Stage detectors also inevitably suffers from robustness bottlenecks.

**Robustness in Region Proposal Network (RPN).** The localization results in the second stage depend on the proposal selected by the localization results of RPN. Therefore, we also analyze the robustness in RPN. The unrectified bounding boxes in the first stage are considered. In Fig. 6, the standard model in the first stage can already give fairly accurate bounding boxes on the clean images. But the validation of $R_{all}$ and $R_{idx}$ indicates that the robust model does not show strong localization robustness at this stage. Under adversarial attacks, the bounding boxes at the correct positions have a

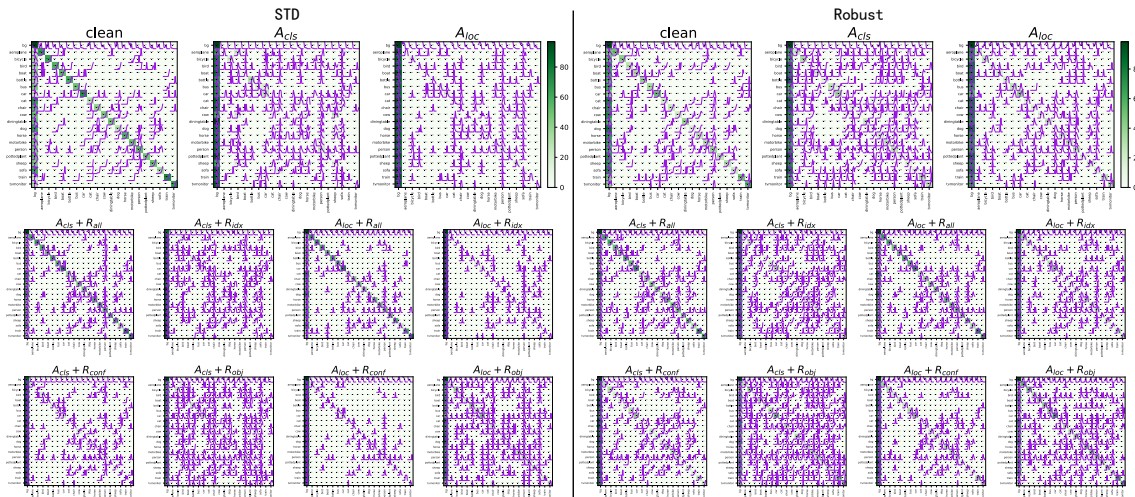

Figure 8: Robustness analyses with *Detection Confusion Matrix* for standard and robust YOLOX.

large localization deviation. Even so, it still has correct objects in the bounding boxes, so the second stage is able to rectify the bounding boxes.

## 4.2 ROBUSTNESS IN ANCHOR-FREE DETECTOR

According to the previous analyses, the robustness bottleneck of object detectors is mainly attributed to the mis-classification of the foreground into the background with poor classification robustness. The localization module has a reliable robustness. So, what if we can decide whether the bounding box is the background at the localization branch? Is the problem of the robustness bottleneck likely to be solved? To verify this hypothesis, we further validate these problems in YOLOX together with its detection robustness.

YOLOX proposes the decouple head which uses regression features in the localization branch to determine whether there is an object inside the bounding box. It multiplies the Sigmoid outputs (scores) of $H_{cls}$ and $H_{obj}$ as the final score instead of Softmax. It has two scores from $H_{cls}$ and $H_{obj}$ and thus has three manners for the ClsAVal validation: $\boldsymbol{R}_{all}$ considers both scores; $\boldsymbol{R}_{obj}$ considers the outputs of $H_{obj}$; $\boldsymbol{R}_{conf}$ replace only the outputs of $H_{cls}$.

Since YOLOX does not directly give the confidence that the object belongs to the background, we set the confidence of the background to: $c_{bg} = 1 - max(H_{cls}(F_{cls}(f(x)) \times H_{obj}(F_{loc}(f(x)))))$. From the results of DCM in Fig. 8, robust YOLOX also has limited robustness performance under adversarial attacks. It still tends to mis-classify objects into the background. This is similar to SSD and Faster RCNN, indicating that placing the module of deciding whether the bounding box contains foreground objects into the localization branch cannot alleviate the robustness bottleneck. According to $\boldsymbol{R}_{all}$, the robust YOLOX can correctly locate objects with robust localization and is hardly affected by the non-robustness of $H_{obj}$.

Since the final confidence of YOLOX consists of both $H_{cls}$ and $H_{obj}$, we also explore which part is dominantly responsible for the limited robustness. In Fig. 8, with results of $\boldsymbol{R}_{all}$ and $\boldsymbol{R}_{idx}$ that the object detector has small conflicts in the localization module with reliable robustness. With results of $\boldsymbol{R}_{obj}$, if the model can correctly distinguish the foreground from the background, it can give the object higher confidence. With results of $\boldsymbol{R}_{conf}$, even if the classification module classifies all the objects correctly, the detector still regards objects as the background finally. The conflict among foreground classes is much weaker than the conflict between foreground and background classes.

## 4.3 ROBUSTNESS IN TRANSFORMER-BASED DETECTOR

We analyze the robustness of Deformable-DETR on MS-COCO Lin et al. (2014). Transformer-based detector has a fixed number of object queries responsible for predicting different objects, but the correspondence between GTs and the bounding box predicted by each object query will change with learning. Here we assume that object queries of the same model are responsible for the same objects under an image with different perturbations. In this way, $\boldsymbol{R}_{all}$ and $\boldsymbol{R}_{idx}$ under different models using their respective results on clean images. Deformable-DETR uses a Sigmoid to predict the category, similar to YOLOX. So we use a similar method to assign the background class confidence: $c_{bg} = 1 - max(H_{cls}(f(x)))$.

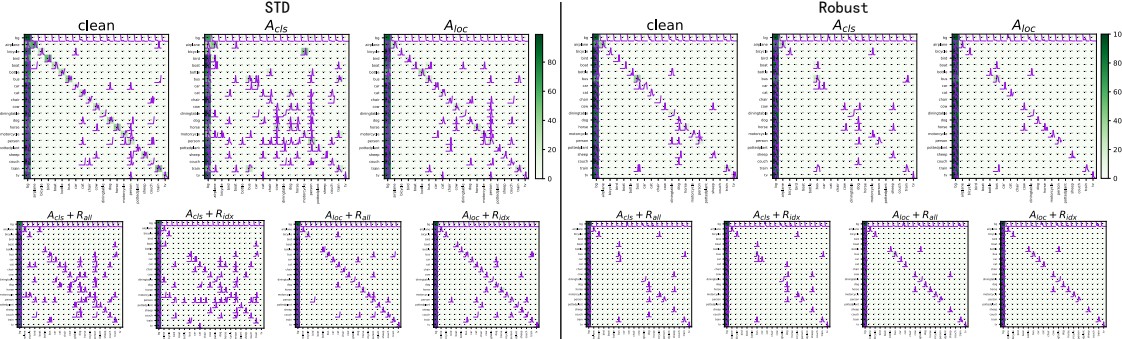

Figure 9: Robustness analyses with *Detection Confusion Matrix* for standard and robust Deformable-DETR models.

We select objects in COCO whose category is the same as the VOC dataset to calculate DCMs. From DCM results in Fig. 9, the standard model not only mis-classifies a large number of foreground objects into the background, but also a lot of confusion exists among foreground categories. Under $R_{all}$, the model identifies numerous objects into the background, which indicates that the standard Transformer-based detector is poor for localization robustness. The bounding boxes at the correct positions has a large deviation, which makes it unable to match the correct true bounding boxes.

DCM of the robust detector on clean images shows that, although it achieve 34.8% mAP, the category confusion is very serious. The robust detector also tends to mis-classify foreground objects as the background under $A_{cls}$ and $A_{loc}$ attacks, and even some objects are not recognized at all. From the results of $R_{all}$ and $R_{idx}$, even if the bounding boxes at the correct positions are given, regardless of what the class scores are, the model still has a poor performance. This indicates that the robustness of the localization module in the Transformer-based detector is as poor as the classification module, and the conflict is also serious. The Transformer-based detector has worse robustness than CNN-based detectors. Its detection robustness bottleneck is not only closely related with the classification but also with the localization.

## 5 SUMMARY AND CONCLUSION

Existing methods present a detection robustness bottleneck: a significant performance decline on clean images and a limited performance on adversarial images. In this paper, we aim to make an initial attempt to comprehensively investigate the adversarial robustness of four seminal object detectors with different architectures, including the one-stage, two-stage, anchor-free, and Transformer-based detectors. With the proposed Detection Confusion Matrix and Classification-Ablative Validation, there are mainly three conclusive analyses:

1) For adversarial attacks, standard and robust detectors tend to mis-classify foreground objects into the background instead of confusion among foreground categories, which is different from the adversarial robustness in image classification.

2) Robust object detectors with different architectures via adversarial training present the robustness bottleneck. For CNN-based detectors, the main reason is that the classification module confuses the foreground and the background so that detectors select the bounding boxes at the wrong positions. The localization module itself has reliable localization robustness. In the Transformer-based detector, its classification robustness and localization robustness are poor.

3) The localization module is highly robust, but determining whether the object belongs to the foreground or the background in the localization branch cannot alleviate the conflict between them. Adversarial attacks for detection models with various structures will seriously confuse the foreground and the background.

To break the detection robustness bottleneck, one crucial research topic is how to alleviate the mis-classification of the foreground objects into the background. Especially, the label-assignment routine for object detection regards both bounding boxes that fully contain background and those that contain partial foreground as background; only those that precisely contain objects are labeled as foreground. Thus, fooling the detector just needs to successfully attack part of the foreground, which may create the learning conflicts in adversarial training. Besides, among those detectors, Transformer-based Deformable-DETR presents inferior robustness, which is worthy of further considerations. Our work is expected to inspire more researches on the adversarial robustness for object detection in the future.

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
