# OpenReview forum: "What's Wrong with the Robustness of Object Detectors?"
_ICLR.cc/2023/Conference — Submitted to ICLR 2023_

### Official Review · Reviewer_K7hu · 2022-10-24

**Confidence:** 4
**Correctness:** 2
**Technical Novelty And Significance:** 2
**Empirical Novelty And Significance:** 2
**Recommendation:** 5

**Clarity, Quality, Novelty And Reproducibility:**

- Since $H_{cls}$ and $H_{loc}$ yield probabilistic confidence and the predicted localization for each bounding box, in Equation (1), $L_{cls}$ requires only $y_{i}$, and $L_{loc}$ requires only $b_{i}$.
The way of expression needs to be explained more clearly. The same goes for Equation (2).

- Figure 2 is not well understood. What are the criteria for dividing into robust and non-robust in the same model? Why is there a difference in performance within the same model? It is not intuitive to paste the performance of (a) into (b), and the caption needs more detail.

- Existing papers come up with novel methods to solve these problems, and it would be persuasive if there was an experiment to see how much these indicators improved.

- The two proposed modules are reproducible and applicable to existing models. Authors also provide reproducible code.

**Strength And Weaknesses:**

Strengths
- Many existing models fail to explain what process they went through when producing incorrect results. This work points to these problems.
- Figure 3 and Figure 4 are sufficient to understand each module.
- Supplementary materials provide code and various experimental results.
- Various analyses of the experimental results suggest the direction of future research in this task.


Weaknesses
- Existing papers also point out this problem and show the need for a new model, which this paper claims is an initial attempt for analysis. A novel motivation is needed for proper analysis.
- Writing is not clear. All paragraphs should be written clearly so that even a first-time reader can understand them.
- I wonder if the object detection methods can be improved with this metric. Isn't it too consequential analysis to use both modules after applying NMS?
- The results can be intuitively confirmed through visualization, but I am not sure that this can be generalized to lead to a specific interpretation.

**Summary Of The Paper:**

To alleviate the detection robustness bottleneck in the future work, this paper investigated the robustness of object detectors.
This work proposes two module: Detection Confusion Matrix (DCM) and Classification-Ablative Validation (ClsAVal).
DCM is used to analyze the confusion of detection results between different categories and their localizations,
and ClsAVal is used for detection robustness analysis.

**Summary Of The Review:**

This paper proposes two modules as indicators for confirming the robustness of existing object detectors for the adversarial attacks.
Experimental results applied to multiple cases show various interpretations in question.
However, these points are not considered novel, and it is hard to see what changes these indicators show when they actually improve.

---

> ### Author Response · Authors · 2022-11-18
> **Response to Reviewer K7hu**
>
> *`Q1:` Existing papers also point out this problem and show the need for a new model, which this paper claims is an initial attempt for analysis. A novel motivation is needed for proper analysis.*
>
> `A1:` In ["Adversarially-aware robust object detector" in ECCV 2022], it just stated that the clean and the adversarial images would have a conflict in the adversarial training, but it did not explain what this conflict is. In this paper, we clearly show that this conflict is between foreground and background through extensive experimental analysis. Adversarial training makes the model tend to predict the foreground as the background. Clarifying the essence of this conflict can bring a stronger insight for further work.
>
> ---
> *`Q2:` Writing is not clear. All paragraphs should be written clearly so that even a first-time reader can understand them.*
>
> `A2:`  Thank you for your suggestions, we will make improvements.
>
> ---
> *`Q3:` I wonder if the object detection methods can be improved with this metric. Isn't it too consequential analysis to use both modules after applying NMS?*
>
> `A3:` DCM can indeed provide some insights for existing object detection methods. Compared with mAP, DCM can analyze the confusion of object detectors among categories and the distribution of bounding box and inter-GT IOU in more detail. DCM can find some bottlenecks of the current detectors and motivate the design of specific methods. For example, in the analysis of Fig.8 for YOLOX, it can be seen that both cats and dogs are more inclined to be predicted as background, but the IOUs of correctly recognized boxes are quite high.
>
> ---
> *`Q4:` The results can be intuitively confirmed through visualization, but I am not sure that this can be generalized to lead to a specific interpretation.*
>
> `A4:` Those interpretations are also provided in the paper based on our quantitative analysis and qualitative validation.
> - Quantitative analysis: We have evaluated the performance of object detectors, including accuracy, IOU, and object scores for each object category. Based on those results, our DCMs are derived for statistical analyses, for example, Fig.5.
> - Qualitative validation: To further verify the robustness of the localization and classification, our clsAVal is conducted, for example in Fig.6.
>
> ---
> *`Q5:` Since $H_{cls}$ and $H_{loc}$ yield probabilistic confidence and the predicted localization for each bounding box, in Equation (1), $L_{cls}$ requires only $y_i$, and $L_{loc}$ requires only $b_i$. The way of expression needs to be explained more clearly. The same goes for Equation (2).*
>
> `A5:` Thanks for your suggestion. We will carefully provide detailed explanations. $y_i$ is the one-hot label for the classification module, and $b_i$ is the parameter of the GT box [x,y,width,height].
>
> ---
> *`Q6:` Figure 2 is not well understood. What are the criteria for dividing into robust and non-robust in the same model? Why is there a difference in performance within the same model? It is not intuitive to paste the performance of (a) into (b), and the caption needs more detail.*
>
> `A6:` The robust and non-robust models are the models obtained by using adversarial training and standard training, respectively. The horizontal coordinate indicates the performance of the model evaluated on the different images (clean or adversarial image).

---

### Official Review · Reviewer_PhPF · 2022-10-25

**Confidence:** 5
**Correctness:** 2
**Technical Novelty And Significance:** 1
**Empirical Novelty And Significance:** 1
**Recommendation:** 1

**Clarity, Quality, Novelty And Reproducibility:**

The writing of this paper is poor. The novelty of it is also minor. I'm not sure if the results can be reproduced.

**Strength And Weaknesses:**

Pros:
+ Adversarial attack in object detection is an important issue. The problem is practical and worth looking into.
+ The authors attempt to figure out which part fails for different models, which is a good attempt.

Cons:
- The paper is difficult to read. There are a lot of issues in writing. Most papers are not cited in the right way. For example, in ''FGSM Goodfellow et al. (2015), PGD Madry et al. (2018), AdvGAN Xiao et al. (2018)', the name of the model should not be messed with the author names. It is suggested that the authors look into the previous ICLR-accepted papers. Also, there are a lot of grammar issues. For example, ''Chen et al. (2018b) have been proposed to attack the state-of-the-art object detectors.' --> Chen et al. have proposed...
- The motivation and previous issues in these models in Introduction have already been discussed in previous papers, e.g. Adversarially-aware robust object detector in ECCV 2022. Fig 1 is also quite similar to Fig.2 in that paper.
- As a paper to investigate different types of detection models, the authors should have a good literature review on object detection models. However, the object detection models are categorised as two-stage, one-stage, anchor-free, and Transformer-based detectors. It's not an accurate classification, e.g., one-stage detectors can be either anchor-free or not. Previous papers have already clearly discussed the critical point that makes the detectors different from each other, e.g. ATSS (Bridging the Gap Between Anchor-based and Anchor-free Detection via Adaptive Training Sample Selection). It is suggested to do a redundant literature review before discussing how the adversarial attack affects the detectors so that the authors can decide how and why these different categories should affect the investigation.
- The main observation that the classification module affects more in the adversarial attack has already been discussed in Section 3.2 of  Adversarially-aware robust object detector in ECCV 2022.
- Section 3.1  is very difficult to understand. The authors designed a very complex method (it looks like it matches the object index between attacked or clean images) to investigate the localization branch independently. It can be simply done by calculating the loss change as proposed in Adversarially-aware robust object detector in ECCV 2022.
- The results after NMS in Fig.3 are very confusing. It doesn't look like NMS has been done, as there are highly overlapped boxes. And the text with the bounding box sits in different locations in the same image, which makes it even more difficult to investigate clearly.
- The authors also report things like ''it is unable to successfully train Deformable-DETR on PASCAL VOCusing the official code.'', which makes me doubt the reproducibility of results in this paper.


**Summary Of The Paper:**

The paper attempts to investigate the problem of why object detection models fail with adversarial attacks. The authors tried to look into the classification and localization branch independently to locate the issue. Also, different types of object detection networks were investigated for this issue. According to experimental results, the authors made some conclusions about which part tasks more mistakes.

**Summary Of The Review:**

The paper investigates an essential and interesting issue. However, due to poor writing, serious issues in clarity and lack of novelty, I tend to reject this paper.

---

> ### Author Response · Authors · 2022-11-18
> **Response to Reviewer PhPF**
>
> *`Q1:` The paper is difficult to read.*
>
> `A1:` Thanks for your review time and comments. There are very few works to investigate the adversarial robustness of existing object detectors, especially including four seminal object detectors. Thus, it is a challenging task to comprehensive this issue. Just because of this, we have carefully and logically organized our paper and elaborated our work. For example, we are well-motivated (Sec. 1), introduce the basic problem defination and the adversarial robustness performance for detection models (Sec. 2), provide a detailed analyses on the detection robustness taking SSD for example (Sec. 3) and then extend to other object detectors to further analyze the detection robustness  (Sec. 4). Pariticularly, we have hightlight our main ideas for analyses using different problems. We believe our paper has been carefully prepared for reading.
>
> ---
> *`Q2:` The motivation and previous issues in these models in Introduction have already been discussed in previous papers.*
>
> `A2:`  In ["Adversarially-aware robust object detector" in ECCV 2022], it just stated that the clean and the adversarial images would have a conflict in the adversarial training, but it did not explain what this conflict is. In this paper, we clearly show that this conflict is between foreground and background through extensive experimental analysis. Adversarial training makes the model tend to predict the foreground as the background. Clarifying the essence of this conflict can bring a significant insight for further work.
>
> ---
> *`Q3:` the literature review on object detection models.*
>
> `A3:`  We did not divide the detectors into four categories without investigation. The reason for dividing the detectors into these four classes is that it is more beneficial for analyzing robustness. We definitely know that one-stage detectors can also be anchor-free. Here we analyze the anchor-free feature separately because in the analyses in the previous sections we have found that the detector tends to predict the foreground as the background after adversarial training. We hypothesize that this may be due to the prior of the anchor, so we try to analyze the anchor-free detector separately. A large intersection of foreground and background anchors during training may bring conflicts. The experiments then demonstrate that the anchor-free detector is not effective in solving this problem either.
>
> ---
> *`Q4:` The main observation that the classification module affects more in the adversarial attack has already been discussed in Section 3.2 of ECCV 2022.*
>
> `A4:` It's not our main observation. The robust object detector will give inaccurate bounding boxes and categories in most cases. And just the claim that the classification module affects more cannot explain the above phenomenon. Our analysis by DCM and ClsAVal demonstrates that the inaccurate bounding boxes are due to the wrong bounding boxes selected by the classification module rather than the poor localization module itself.
>
> ---
> *`Q5:` The authors designed a very complex method to investigate the localization branch independently. It can be simply done by calculating the loss change as proposed in ECCV 2022.*
>
> `A5:`  We need a quantitative metric to individually evaluate how well the model's localization module performs. Just like we do not evaluate the performance of a detector by loss, but using mAP. The calculation of mAP is obviously much more complicated than loss, so why should we use mAP instead of loss? Loss change can only roughly judge that the classification module performs a bit worse, but cannot evaluate the performance of the localization module. After eliminating the effect of the classification module by our method, it can be seen that the performance of the localization module itself is quite robust, and even better than standard on YOLOX. These tools and conclusions can provide the basis for further works.
>
> ---
> *`Q6:` The results after NMS in Fig.3 are very confusing. It doesn't look like NMS has been done, as there are highly overlapped boxes. And the text with the bounding box sits in different locations in the same image, which makes it even more difficult to investigate clearly.*
>
> `A6:` This is just a schematic, the text has been adjusted to show each part clearly. The results of NMS will depend on the selection of thresholds, and selecting different thresholds will show different performance.
>
> ---
> *`Q7:` The authors also report things like ''it is unable to successfully train Deformable-DETR on PASCAL VOCusing the official code.'', which makes me doubt the reproducibility of results in this paper.*
>
> `A7:` The poor peformance of Deformable-DETR on PASCAL VOC is widely demonstrated and known. This issue is also discussed in the official **github issues** and has not been well solved. Thus, it is trivial and meanless to evaluate this model on this dataset. This has nothing with our reproducibility.

---

### Official Review · Reviewer_dqA6 · 2022-10-28

**Confidence:** 3
**Correctness:** 3
**Technical Novelty And Significance:** 2
**Empirical Novelty And Significance:** 2
**Recommendation:** 5

**Clarity, Quality, Novelty And Reproducibility:**

### Clarity
Can be improved. See the weakness above.

### Quality
Good

### Novelty
Not strong enough. See the weakness above.

### Reproducibility
Should be reproducible.

**Strength And Weaknesses:**

### Strength
* The paper analyzes a large group of detector types, and analyzes object detector with and without adversarial training.
* The paper draws some clear conclusion, which may be helpful for future works on improving the robustness of object detectors.

### Weaknesses
* The novelty is limited and the findings are not super surprising that the adversarial attack is mostly making the classification wrong.
* I feel the conclusion made in section 3.2 that "The adversarial attack enforces the standard detection model to easily mis-classify foreground objects into the background" may depend on the attack method. For example, in figure 1 of [a], the dogs are detected to be "train" with high confidence score.
* Conclusions in section 3.2 (a) are mostly drawn from figure 6 which is qualitative analysis. More careful quantitative analysis is required.
* The clarity of the paper can be improved. For example:
  * In equation 1, the classification loss is only calculated on $y_i$ and $\mathbf{b}_i$. I feel this is incorrect since background classification should also be considered in loss?
  * In section 3.1 ClsAVal, should we emphasize that $M$ and $M^R$ have the same structure (e.g. $M^R$ is an adversarially trained counterpart of $M$), so that the indices $\it{idx}$ from one detector can be applied to the other? This is not super clear in that paragraph so it's a bit confusing.
  * It would be better to explain a bit the details of "$R_{idx}(A_{cls})$" etc in the caption of figure 6.

[a] Adversarial Examples for Semantic Segmentation and Object Detection

**Summary Of The Paper:**

In this paper, authors investigates the failure reasons of object detectors under adversarial attacks. Authors propose two analysing strategies, named DCM and ClsAVal, to understand the classification confusions and the contribution of localization failures. Authors analyzed 4 types of object detectors (single-stage, two-stage, anchor-free and transformer query based), and focused on the single-shot detectors with their analysing strategies. Some conclusions were made such as the classification confusion is usually the more serious root cause of detection failures.

**Summary Of The Review:**

The paper provides adversarial robustness analysis on a large group of object detectors. However, the novelty is not strong enough and there are still some concerns in the current version, thus I lean towards weak reject at the current stage.

---

> ### Author Response · Authors · 2022-11-18
> **Response to Reviewer dqA6**
>
> *`Q1:` The novelty is limited and the findings are not super surprising that the adversarial attack is mostly making the classification wrong.*
>
> **`A1:`** Thanks for the question. However, we respectfully disagree with your comment. Although adversarial robustness is widely-explored in image classification, few works are devoted to robustness for object detection, especially how to defend those attacks for object detection, which are almost blind in existing works.  Thus, rather than proposing a specific solution for robust object detectors as usual, we specifically aim to make an initial attempt to investigate the robustness of object detectors systematically and comprehensively, paving a way for future works that are inspired to essentially mitigate the detection robustness bottleneck. In a nutshell, our paper is not only well-motivated to explore the robustness of object detection, but also has proposed two novel detection metrics and made a series of important investigations and insightful conclusions on adversarial robustness for object detectors. Besides, the comment that the adversarial attack is mostly making the classification wrong is NOT our finding in our work. Next, we will elaborately explain your concerns.
>
> - Novelty and Contribution:
>     - We make an early attempt to investigate the adversarial robustness of existing object detectors (including **four** seminal detectors, including two-state, one-stage, anchor-free, and transformer-based detection models), which are greatly threaded by adversarial attacks.
>     - We propose a novel Detection Confusion Matrix (DCM) in object detection to analyze the classification and localization in detail and a novel Classification-Ablative Validation for detection robustness analysis.
>     - With our extensive analyses, four seminal detectors are consistently agonized with similar vulnerability issues: e.g., mis-classification of foreground into background, reliable localization robustness in CNN-based detectors, and poor localization in Transformer-based detectors.
>
> - Finding and Conclusion:
>     - With our comprehensive evaluation, our three conclusive findings (See Conclusion section) convincingly diagnose what's wrong with adversarial detection robustness in existing detectors, which is almost blind in existing works. Especially, this paper paves a way to break the detection robustness bottleneck and motivates us to rethink existing object detectors. For example, the conventional label-assignment routine for object detection is problematic, which aggregates the mis-classification of objects into the background.
> ---
> *`Q2:` The conclusion made in section 3.2 that "The adversarial attack enforces the standard detection model to easily mis-classify foreground objects into the background" may depend on the attack method.*
>
> `A2:` From our analysis of the entire test set using DCM (in Fig.5, 7, 8), it can be observed that most of the foreground bounding boxes are incorrectly classified into the background. It is true that a few bounding boxes are classified into other categories, but in statistical terms, most of the bounding boxes are classified into the background.
>
> ---
> *`Q3:` Conclusions in section 3.2 (a) are mostly drawn from figure 6 which is qualitative analysis.*
> `A3:` We have drawn conclusions in section 3.2 from two analytical aspects, namely, quantitative analysis and qualitative validation.
> - Quantitative analysis: We have evaluated the performance of object detectors, including accuracy, IOU, and object scores for each object category. Based on those results, our DCMs are derived for statistical analyses, shown in Fig.5.
> - Qualitative validation: To further verify the robustness of the localization and classification, our clsAVal is conducted, as shown in Fig.6.
>
> ---
> `Q4:` The clarity of the paper can be improved.
>
> *`A4:`*
> - *In equation 1, background classification is not considered in loss.* In conventional object detection formulations, only object proposals that are predicted to match the ground-truth object boxes are used to compute the loss. For example, the loss function is also defined in existing SSD ["SSD: single shot multibox detector", ECCV2016]. Besides, it is also used for adversarial attacks in the MTD model ["Towards adversarially robust object detection", ICCV2019]. Thus, we follow the same formulation in Equ.1.
>
> - *$M$ and $M^R$ in section 3.1 ClsAVal.* Thanks for your suggestion. $M^R$ and $M$ have the same structure and are just trained under different settings. For example, when evaluating the robust model obtained from adversarial training as $M$,  the standard model can be used as $M^R$. Because the classification prediction results of the standard model are relatively reliable.
>
> - *"$R_{idx}(A_{cls})$" etc in the caption of figure 6.* $R_{idx}(A_{cls})$ and $R_{idx}(A_{loc})$ denote using $R_{idx}$ to analyze the adversarial attacks against classification and localization loss, respectively.

---

### Decision · Program_Chairs · 2023-01-20

**Decision:**

Reject

**Justification For Why Not Higher Score:**

This is a clear rejection.

**Justification For Why Not Lower Score:**

N/A

**Metareview: Summary, Strengths And Weaknesses:**

The paper presents a comprehensive study on the susceptibility of different object-detection pipelines to adversarial images. While the study in itself appears to be well-executed, it does not appear to lead to any meaningful new insights. In particular, the paper concludes that detection pipelines confuse the foreground and background class when confronted with adversarial images — this is exactly what the large literature on adversarial image generation designed adversarial images to do, so this conclusion is hardly surprising. As the study described in the paper does not lead to major new observations or learnings, its value to the ICLR community is limited.

**Summary Of Ac-Reviewer Meeting:**

N/A